# A Multi-view Latent Space Learning Framework via Adaptive Graph Embedding

## Abstract

In this paper, a new approach to multi-view subspace learning is proposed and termed as multi-view latent space learning via adaptive graph embedding (MvSLGE), which learns a latent representation from all view features. Unlike most existing multi-view latent space learning methods that only encode the complementary information into the latent representation, MvSLGE adaptively learn an adjacent graph that well characterizes similarity between samples to further regularize the latent representation. To extract the neighborhood information from multi-view features, we propose a novel strategy that constructs one graph for each view, and then the learned graph is approximately designed as a centroid of these graphs of different views with different weights. Therefore, the constructed latent representation not only incorporates the complementary information of features from multiple views but also encodes the similarity between samples. The proposed MvSLGE can be solved by the augmented Lagrangian multiplier with alternating direction minimization (ALM-ADM) algorithm. Plenty of experiments demonstrate the superiority of MvSLGE on a variety of datasets.

## 1 Introduction

Most of the data used in many fields can usually be represented by different modalities or different types of features. For example, in the computer vision domain, one color image can be described by multiple features, such as LBP, SIFT, HOG, Gist, etc Wang et al. (2017). Since an individual view can not contain sufficient information to comprehensively describe all samples, multiple views features are needed for various applications. However, different features that describe one sample are constructed based on different purposes, which leads a large gap between views. It is therefore important to effectively utilize information from multi-view features to the fullest extent possible for depicting the objects better and improving various performances.

Recently, various multi-view algorithms have been proposed to solve this problem. Among all these algorithms, multi-view subspace learning is a very important direction Jia et al. (2021); Lin et al. (2023). One of the most famous algorithms is canonical correlation analysis (CCA) Hotelling (1992) and its kernel extension kernel canonical correlation analysis (KCCA) Lai & Fyfe (2000). CCA aims to seek a projection matrix for each view to project them onto one common subspace by maximizing the correlation of individual views. However, they don't account for the independent parts of the views. Therefore, these methods totally fail to represent them or mix them with the information shared by all views. Jia et al. (2010) consider the latent space as two parts: one common part to all views and the other one for each individual view, which can account for the independencies and dependencies between views simultaneously. However, there exist some limitations to these methods. The main limitations are losing information in the procedure of learning from a single view feature which contains insufficient information, or noise in the original single view feature. In order to solve these limitations, Xu et al. (2015) propose a novel multi-view subspace learning algorithm termed as multi-view intact space learning (MISL), which seeks to construct one common intact latent space from multi-view features. Besides, to improve the robustness of the model, MISL measures the reconstruction error of each view by Cauchy loss. Nevertheless, MISL hasn't considered geometric structure information in the constructed latent representation. Since the features from different views usually lie in different distributions, how to explore the underlying manifold structure from multi-view features is still challenging.

To deal with this problem, we propose a multi-view latent space learning via adaptive graph embedding (MvSLGE). The proposed MvSLGE aims to fuse the multi-view features to construct one common latent representation which can effectively incorporate complementary information of heterogeneous property features. Moreover, to encode the local geometry structure of samples, MvSLGE builds a similarity graph to regularize the latent representation. This graph is constructed from the learned latent representation and original multi-view features. Additionally, considering that different views make different contributions, we futher propose an auto-weighted scheme to learn an appropriate weight for each view. Finally, we adopt the ALM-ADM to solve the proposed MvSLGE.

## 2 RELATED WORK

With increasing multi-view data, numerous algorithms for multi-view learning have shown remarkable success in various real-world applications Jia et al. (2021); Lin et al. (2023); Yan et al. (2021); Liang et al. (2022). Most multi-view learning methods can be classified into three groups: co-training, multiple kernel learning (MKL), and multi-view subspace learning. Co-training Blum & Mitchell (1998) is a famous semi-supervised learning model for dealing with multi-view data. The co-training seeks to maximize consensus between two different views. There are many variants, such as co-regularization Kumar et al. (2011) and co-EM Nigam & Ghani (2000), have been developed. These variants are applied to multi-view dimension reduction Zhang et al. (2017), multi-view metric learning Wang et al. (2017); Zhang et al. (2018) and multi-view clustering Kumar et al. (2011); Cao et al. (2015).

MKL is very popular since it provides a convenient way of combining information from multi-view data. MKL considers the different kernels as different views and combines kernels to enhance the performance. In Lanckriet et al. (2004), MKL is reformulated into a semi-definite programming problem, which can be solved easily. In Sonnenburg et al. (2006) MKL is developed and formulated as a more efficient semi-infinite linear program, which can be utilized to deal with large-scale data. Szafranski et al. (2010) and Xu et al. (2010) introduce group-LASSO into MKL and propose a novel MKL approach, which can well describe group structure for multi-view data and obtain generalization bounds.

Multi-view subspace learning is also famous recently, which aims to discover one shared subspace across all different views Wu et al. (2019); Wang et al. (2015); Zhao et al. (2018). CCA Hotelling (1992), and its kernel extension Lai & Fyfe (2000) KCCA are the earliest typical multi-view subspace learning algorithms. In order to extend the traditional CCA to multiple views case, Rupnik & Shawe-Taylor (2010) propose multi-view CCA. Motivated by CCA and Fisher discriminant analysis (FDA), Diethe et al. (2008) introduce the regularized two-view equivalent of FDA and its corresponding dual. In contrast to CCA, this generalization fully considers the label information in their algorithm. In Sharma et al. (2012) a multi-view subspace learning framework is proposed as generalized multiview analysis (GMA), which can be considered as an extension of traditional CCA algorithm. Deep canonical correlation analysis (DCCA) Andrew et al. (2013) exploits neural networks framework to explore the nonlinear correlation between different views. In the recent decade, multi-view subspace clustering has become more popular. Based on the complementary principle, Cao et al. (2015) propose a multi-view clustering approach termed as diversity-induced multi-view subspace clustering (DiMSC), in which Hilbert-Schmidt independence criterion (HSIC) is adopted for measuring the dependencies between views. Then, by minimizing the dependencies of the views, DiMSC effectively extracts more incompatible information from multiple views features. Zong et al. (2017) incorporate matrix factorization and multi-manifold regularization into a framework to propose a multi-view subspace learning model for multi-view data clustering, which focuses on learning the locally manifold structure of samples from multi-view features.

## 3 MULTI-VIEW LATENT SPACE LEARNING VIA ADAPTIVE GRAPH EMBEDDING

### 3.1 THE CONSTRUCTION OF MVSLGE

Given $N$ samples with $m$ views features $\left\{ \boldsymbol{Z}^{(v)} \middle| \boldsymbol{Z}^{(v)} \in \mathbb{R}^{D_v \times N} \right\}_{v=1}^{m}$, where $D_v$ is the dimensionality of features from the $v$th view, our method aims to seek a latent space in which multi-

view feature of one sample can be reconstructed by one common latent representation $\boldsymbol{X} = [\boldsymbol{x}_1, \boldsymbol{x}_2, ..., \boldsymbol{x}_N] \in \mathbb{R}^{D \times N}$ by $m$ linear view generation functions $\{\boldsymbol{P}_v\}_{v=1}^m$, where $D$ denotes the dimensionality of the latent representation. With noise in features, we have

$$\boldsymbol{z}_i^{(v)} = \boldsymbol{P}_v \boldsymbol{x_i} + \boldsymbol{e}_i^{(v)}, \tag{1}$$

where $\boldsymbol{e}_i^{(v)}$ denotes the noise of the $i$th sample of the $v$th feature. Within the empirical risk minimization framework, the objective function Xu et al. (2015) becomes

$$\min_{\boldsymbol{X},\boldsymbol{P},\boldsymbol{E}} \quad \mathcal{L}_{\mathrm{M}}\left(\boldsymbol{E}\right),$$
$$s.t. \quad \boldsymbol{E} = \boldsymbol{Z} - \boldsymbol{P}\boldsymbol{X}, \tag{2}$$

where $\boldsymbol{Z} = [\boldsymbol{Z}_1; \boldsymbol{Z}_2; ...; \boldsymbol{Z}_m]$ and $\boldsymbol{P} = [\boldsymbol{P}_1; \boldsymbol{P}_2; ...; \boldsymbol{P}_m]$ are original multi-view features and corresponding learned linear view generation functions, and $\mathcal{L}_{\mathrm{M}}$ is the error measurement. By Equation 2, we can encode the complementary information of multi-view features into one common latent representation $\boldsymbol{X}$ automatically. Thus, the learned latent representation can comprehensively depict a sample more than every single view.

To enhance the robustness of the model, we adopt $l_{2,1}$-norm $\left\|\cdot\right\|_{2,1}$ as the measurement of the reconstruction error for each view. The $l_{2,1}$-norm of a matrix $\boldsymbol{B} \in \mathbb{R}^{l \times k}$ is defined as: $\left\|\boldsymbol{B}\right\|_{2,1} = \sum_{j=1}^k \sqrt{\sum_{i=1}^l \boldsymbol{B}_{ij}^2}$. Based on the effect of $l_1$-norm, we can find that minimizing $l_{2,1}$-norm of the matrix makes the columns with small 2-norm to be zero, which makes it robust to outliers. The objective function for minimizing the reconstruction error in our model is

$$\min_{\boldsymbol{X},\boldsymbol{\mathcal{P}},\boldsymbol{\mathcal{E}}} \quad \frac{1}{mN} \left\|\boldsymbol{E}\right\|_{2,1} + \lambda \left\|\boldsymbol{X}\right\|_F^2,$$
$$s.t. \quad \boldsymbol{E} = \boldsymbol{Z} - \boldsymbol{P}\boldsymbol{X}, \quad \boldsymbol{P}\boldsymbol{P}^T = \boldsymbol{I}, \tag{3}$$

where we set $\lambda > 0$, and employ a regularization term to penalize the latent representations $\boldsymbol{X}$ and a constraint on each projection matrix $\boldsymbol{P}_v$.

Laplacian embedding is a famous technique in the manifold learning domain, which aims to explore the local manifold structure of data. Suppose $\boldsymbol{S} \in \mathbb{R}^{N \times N}$ is an affinity matrix of a graph with $N$ nodes. The Laplacian embedding technique intends to compute the embedding of all data points by preserving the neighbor relationship of them as much as possible Belkin & Niyogi (2001). We suppose the latent representations are the embedding coordinates of $N$ sample points. Then we have

$$\min_{\boldsymbol{X}} \quad \sum_{i,j=1}^N \left\|\boldsymbol{x}_i - \boldsymbol{x}_j\right\|_2^2 s_{ij}, \tag{4}$$

where $s_{ij}$ usually characterizes the similarity between $\boldsymbol{x}_i$ and $\boldsymbol{x}_j$. In the manifold learning domain, the affinity matrix is usually predefined, which is not flexible and affects the performance of the algorithm. So we aim to adaptively construct the affinity relationship which can characterize the local manifold structure of samples.

To extract the local geometry structure from multi-view features, we build a graph $\boldsymbol{S}^{(v)}$ for each view as Belkin & Niyogi (2001). Then we aim to construct the integrated affinity matrix $\boldsymbol{S}$ that can optimally integrate the affinity information of data by the pre-defined graphs $\boldsymbol{S}^{(v)}$ of all views. To achieve this goal, we incorporate the graph learning and latent space learning into one framework

$$\min_{\boldsymbol{X},\boldsymbol{P},\boldsymbol{E},\boldsymbol{S}} \quad \frac{1}{mN} \left\|\boldsymbol{E}\right\|_{2,1} + \lambda \left\|\boldsymbol{X}\right\|_F^2 + \eta \sum_{i,j=1}^N \left\|\boldsymbol{x}_i - \boldsymbol{x}_j\right\|_2^2 s_{ij} + \zeta \sum_{v=1}^m \frac{1}{m} \left\|\boldsymbol{S} - \boldsymbol{S}^{(v)}\right\|_F^2,$$
$$s.t. \quad \boldsymbol{E} = \boldsymbol{Z} - \boldsymbol{P}\boldsymbol{X}, \quad \boldsymbol{P}\boldsymbol{P}^T = \boldsymbol{I}, \quad s_{ij} > 0 \quad \text{and} \quad \sum_j s_{ij} = 1, \tag{5}$$

where $\zeta > 0$. Moreover, we further consider that different views usually have different contributions for learning the graph. In order to more appropriately integrate the affinity relationship of all views,

we introduce a collection of non-negative weights $\boldsymbol{a} = [a_1, a_2, ..., a_m]$ into graph learning procedure

$$\min_{\boldsymbol{S},\boldsymbol{a}} \quad \sum_{v=1}^{m} a_v \left\| \boldsymbol{S} - \boldsymbol{S}^{(v)} \right\|_F^2 + \gamma \|\boldsymbol{a}\|_2^2,$$

$$s.t. \quad s_{ij} \geq 0 \quad \text{and} \quad \sum_j s_{ij} = 1, \quad \sum_{v=1}^{m} a_v = 1, \quad a_v \geq 0, \tag{6}$$

where $\gamma > 0$ is a trade-off between the two terms above. The second term in Equation 6 is employed to smoothen the distribution of $a$, which can make all views contribute to the procedure of graph learning. Because without the second term, the weight of one view will be assigned to 1, and others will be 0. Moreover, the solution of minimizing $\|\boldsymbol{a}\|_2^2$ with respect to $\sum_v a_v = 1$ is $a_v = \frac{1}{m}$. Therefore, the second term also makes the weights non-sparse, which can promote all views to participate in the process of learning.

Combining all the above insights, we can finally construct the objective function of MvSLGE as

$$\min_{\boldsymbol{X},\boldsymbol{P},\boldsymbol{E},\boldsymbol{S},\boldsymbol{a}} \quad \frac{1}{mN} \|\boldsymbol{E}\|_{2,1} + \lambda \|\boldsymbol{X}\|_F^2 + \eta \sum_{i,j=1}^{N} \|\boldsymbol{x}_i - \boldsymbol{x}_j\|_2^2 s_{ij} + \zeta \sum_{v=1}^{m} a_v \left\| \boldsymbol{S} - \boldsymbol{S}^{(v)} \right\|_F^2 + \gamma \|\boldsymbol{a}\|_2^2,$$

$$s.t. \quad \boldsymbol{E} = \boldsymbol{Z} - \boldsymbol{P}\boldsymbol{X}, \quad \boldsymbol{P}\boldsymbol{P}^T = \boldsymbol{I}, \quad \sum_{v=1}^{m} a_v = 1, \quad a_v \geq 0, \quad s_{ij} \geq 0 \quad \text{and} \quad \sum_j s_{ij} = 1, \tag{7}$$

where $\lambda$, $\eta$, $\zeta$, and $\gamma$ are non-negative hyperparameters that control the trade-off between four terms in Equation 7.

## 3.2 OPTIMIZATION PROCEDURE OF MVSLGE

In this section, we illustrate the details of the optimization algorithm of MvSLGE. It can be easily found that for all the problems of variables, Equation 7 is not convex, and directly solving the problem is not possible. So we employ the ALM algorithm and divide the original optimization problem into four subproblems which can be efficiently solved by the ADM strategy. Firstly, we formulate the Lagrangian function of the objective function in Equation 7 as follows

$$\mathcal{L}(\boldsymbol{E}, \boldsymbol{X}, \boldsymbol{P}, \boldsymbol{S}, \boldsymbol{a}) = \frac{1}{mN} \|\boldsymbol{E}\|_{2,1} + \lambda \|\boldsymbol{X}\|_F^2 + \eta \sum_{i,j=1}^{N} \|\boldsymbol{x}_i - \boldsymbol{x}_j\|_2^2 s_{ij}$$

$$+ \zeta \sum_{v=1}^{m} a_v \left\| \boldsymbol{S} - \boldsymbol{S}^{(v)} \right\|_F^2 + \gamma \|\boldsymbol{a}\|_2^2 + \mathcal{G}(\boldsymbol{J}, \boldsymbol{Z} - \boldsymbol{P}\boldsymbol{X} - \boldsymbol{E}), \tag{8}$$

$$s.t. \quad \boldsymbol{P}\boldsymbol{P}^T = \boldsymbol{I}, \quad \sum_{v=1}^{m} a_v = 1, \quad s_{ij} \geq 0 \quad \text{and} \quad \sum_j s_{ij} = 1,$$

The $\mathcal{G}(\cdot, \cdot)$ is a function of matrix and defined as: $\mathcal{G}(\boldsymbol{A}, \boldsymbol{B}) = \frac{\mu}{2} \|\boldsymbol{B}\|_F^2 + tr(\boldsymbol{A}^T \boldsymbol{B})$. $\boldsymbol{J}_v$ is a Lagrangian multiplier and $\mu > 0$ is a penalty scalar. We minimize $\mathcal{L}$ for one variable, while other variables are fixed. All the subproblems are illustrated as follows.

(1). $\boldsymbol{P}$-subproblem: To update the generalized function $\boldsymbol{P}$, we fix $\boldsymbol{X}, \boldsymbol{E}, \boldsymbol{a}$ and $\boldsymbol{S}$, and reformulate the objective function for each $\boldsymbol{P}_v$ as

$$\boldsymbol{P}^* = \arg\min_{\boldsymbol{P}} \mathcal{G}(\boldsymbol{J}, \boldsymbol{Z} - \boldsymbol{P}\boldsymbol{X} - \boldsymbol{E})$$

$$= \arg\min_{\boldsymbol{P}} \frac{\mu}{2} \left\| (\boldsymbol{Z} - \boldsymbol{E} + \boldsymbol{J}/\mu)^T - \boldsymbol{X}^T \boldsymbol{P}^T \right\|_F^2 \tag{9}$$

$$s.t. \quad \boldsymbol{P}\boldsymbol{P}^T = \boldsymbol{I}.$$

By some algebra operation, Equation 9 can be further reformulated as

$$\min_{\boldsymbol{P}} \quad tr(\boldsymbol{P}\boldsymbol{X}\boldsymbol{F}^T)$$

$$s.t \quad \boldsymbol{P}\boldsymbol{P}^T = \boldsymbol{I}, \tag{10}$$

where $\boldsymbol{F} = \boldsymbol{Z} - \boldsymbol{E} + \boldsymbol{J}/\mu$.

We can easily optimize Equation 10 based on one famous theorem from linear algebra Gao et al. (2018), which can be illustrated as

$$\boldsymbol{P} = \boldsymbol{W}\boldsymbol{U}^T, \tag{11}$$

where $\boldsymbol{W}$ and $\boldsymbol{U}$ are SVD decomposition of the matrix: $\boldsymbol{X}\boldsymbol{F}^T$, i.e. $\boldsymbol{X}\boldsymbol{F}^T = \boldsymbol{U}\boldsymbol{\Sigma}\boldsymbol{W}^T$.

(2). $\boldsymbol{X}$-subproblem: In the step of updating the latent representation $\boldsymbol{X}$, we fix $\boldsymbol{P}$, $\boldsymbol{S}$, $\boldsymbol{a}$ and $\boldsymbol{E}$ and solve the following minimization problem

$$
\begin{aligned}
\boldsymbol{X}^* &= \arg\min_{\boldsymbol{X}} \eta \sum_{i,j=1}^{N} \|\boldsymbol{x}_i - \boldsymbol{x}_j\|_2^2 \, s_{ij} + \mathcal{G}\left(\boldsymbol{J}, \boldsymbol{Z} - \boldsymbol{P}\boldsymbol{X} - \boldsymbol{E}\right) \\
&= \arg\min_{\boldsymbol{X}} \eta\, tr\left(\boldsymbol{X}\boldsymbol{L}_{\boldsymbol{S}^q}\boldsymbol{X}^T\right) + \frac{\mu}{2} \|\boldsymbol{Z} - \boldsymbol{P}\boldsymbol{X} - \boldsymbol{E} + \boldsymbol{J}/\mu\|_F^2,
\end{aligned} \tag{12}
$$

where $\boldsymbol{L}_{\boldsymbol{S}^q}$ is a graph Laplacian of affinity matrix $\boldsymbol{S}^q$ with $\boldsymbol{S}_{ij}^q = (s_{ij} + s_{ji})/2$. We can find that the optimization problem in Equation 12 is an unconstrained optimization problem. Therefore, the optimal latent representation $\boldsymbol{X}$ can be obtained by setting the gradient of the function in Equation 12 with respect to $\boldsymbol{X}$ as $\boldsymbol{0}$

$$\boldsymbol{M}\boldsymbol{X} + \boldsymbol{X}\boldsymbol{K} = \boldsymbol{O}, \tag{13}$$

where $\boldsymbol{M} = \mu\boldsymbol{P}^T\boldsymbol{P}$, $\boldsymbol{O} = \mu\boldsymbol{P}^T\left(\boldsymbol{Z} - \boldsymbol{E} + \boldsymbol{Y}/\mu\right)$, and $\boldsymbol{K} = 2\eta\boldsymbol{L}_{\boldsymbol{S}^q}$. Equation 13 is the Sylvester equation, which has a unique solution if $\boldsymbol{M}$ and $-\boldsymbol{K}$ have no one eigenvalue in common.

The steps of updating $\boldsymbol{S}$, $\boldsymbol{a}$ and $\boldsymbol{E}$, and the details of processing test sample are demonstrated in the section 1 of supplemental material. Then we further summarize the procedure of MvSLGE in the section 2 of supplemental material. We iteratively alternate the 6 steps until the objective function Equation 8 satisfies convergence conditions. Moreover, we analyze the complexity of the methods in the section 3 of supplemental material.

## 4 EXPERIMENTS

### 4.1 DATASETS AND COMPARING METHODS

To comprehensively evaluating the proposed MvSLGE, 7 famous datasets are employed in the experiments including 3Sources, Cora, Outdoor Scene, Holidays, Caltech 101-7, MSRC-v1, and COIL-20. Among these 7 datasets, 3Sources and Cora are multi-view text datasets, Outdoor Scene, Holidays, Caltech 101-7, MSRC-v1 and COIL-20 are image datasets. The details of the employed datasets are summarized in the section 4 of supplemental material.

We compare MvSLGE with 6 typical multi-view subspace learning methods to evaluate the effectiveness: 1. canonical correlation analysis (CCA) Rupnik & Shawe-Taylor (2010), 2. multiview spectral embedding (MSE) Xia et al. (2010), 3. factorized latent spaces with structured sparsity (FLSSS) Jia et al. (2010), 4. multi-view intact space learning (MISL) Xu et al. (2015), 5. unsupervised multi-view manifold learning with locality alignment (U-MVML-LA) Zhao et al. (2018), 6. multi-view dimensionality co-reduction (McDR) Zhang et al. (2017). The hyperparameters of each method are set by the 5-fold cross-validation.

### 4.2 IMAGE RETRIEVAL

In this section, we conduct image retrieval experiments on Outdoor Scene and Holidays datasets. The outdoor Scene dataset consists of 2688 color images with 8 categories. For each category, 10 images are randomly selected as the query images, while other images are galleries. For each image, 4 views of features are extracted for our experiments: GIST, Color Moment, HoG, and LBP. Then the $L_1$ distance is adopted to measure the similarity. All the algorithms are conducted 10 times with different queries, and the retrieval results are shown in Figure 1.

For the Holidays dataset, 1 image of each category is selected as the query images, and others are assigned as galleries. We exploit the same multi-view features with Outdoor Scene dataset for each image in this experiment. We adopt the $L_1$ distance to measure the similarity. The retrieval results are shown in Table 1.

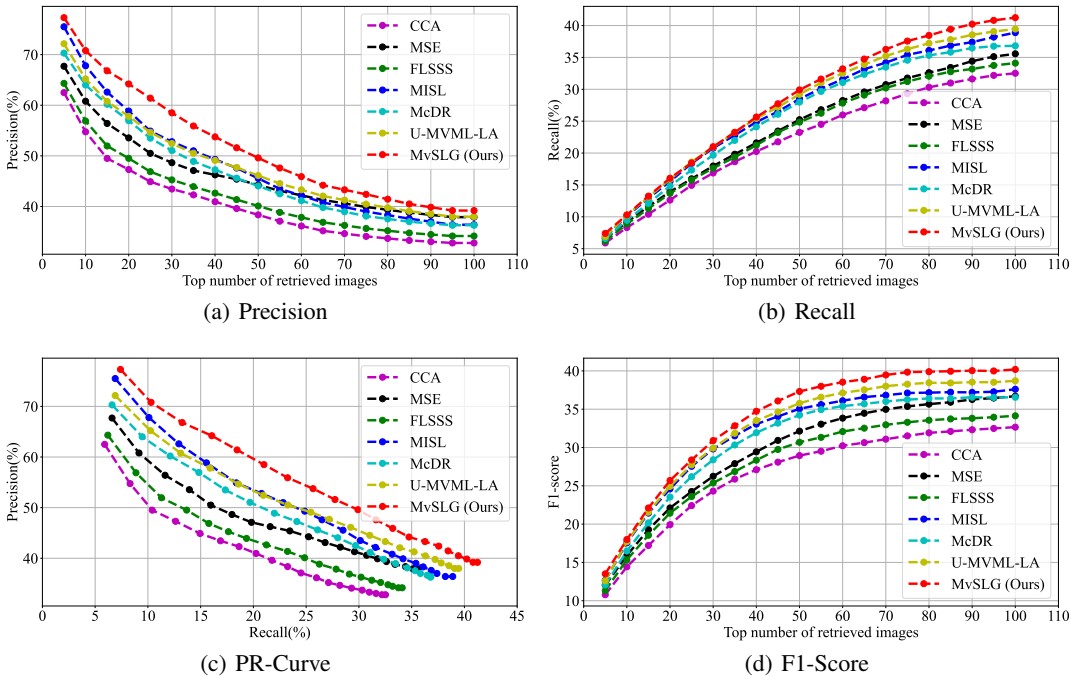

(a) Precision

(b) Recall

(c) PR-Curve

(d) F1-Score

Figure 1: The performance of different algorithms on Outdoor Scene dataset.

Table 1: The performance of different algorithms on the Holidays dataset.

| Method | Criteria | | | |
|---|---|---|---|---|
| | Precision | Recall | MAP | F1-score |
| CCA | 77.15 | 58.12 | 83.13 | 66.28 |
| MSE | 77.65 | 59.42 | 83.65 | 67.32 |
| FLSSS | 76.25 | 57.73 | 82.35 | 65.70 |
| MISL | 78.03 | 60.12 | 83.94 | 67.90 |
| McDR | 77.69 | 60.01 | 84.03 | 67.70 |
| U-MVML-LA | 77.96 | 60.23 | 83.87 | 67.94 |
| **MvSLGE (Ours)** | **79.02** | **61.53** | **85.85** | **69.76** |

From the results Figure 1-Figure 2 and Table 1, We can see that the proposed MvSLGE obtains the best performances. U-MVML-LA utilizes the locality alignment to improve the clustering effect of latent representation and outperforms MISL. However, the graph it adopted to regularize the latent representations is pre-defined. And the graph utilized in MvSLGE for embedding the latent representations is adaptively learned, which is more flexible. Therefore, MvSLGE can achieve better performance.

## 4.3 CLASSIFICATION EXPERIMENTS

For all classification experiments on each dataset, we select 10 train-test splits randomly and conduct 20 test runs for each split. After obtaining the representations, we adopt the SVM classifier to classify testing samples and report the averaged performance.

We first conduct text classification experiments on 2 text datasets 3Sources and Cora. Table 2 shows the averaged values with different dimensionalities respectively. We can easily find from Table 2 that MvSLGE obtains the best performance in most situations. Although MISL can achieve excellent performance by efficiently encoding the complementary information from multi-view features, it hasn't exploited local manifold information. U-MVML-LA considers the local information in its framework, but it constructs the graph artificially. Therefore, MvSLGE is a more effective

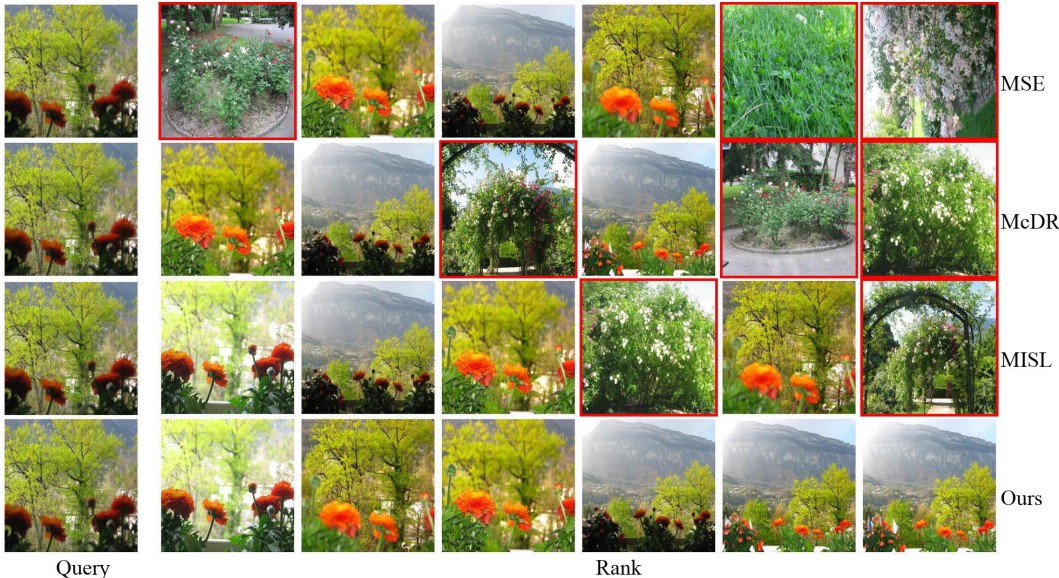

Figure 2: A retrieval example on the Holidays dataset, in which our MvSLG obtains the best results. The left-most image in each row is the query image, and the wrong images are outlined in red.

Table 2: Classification Accuracies (%) with different dimensionality on 3Sources and Cora datasets.

| Method | Dim=10 | | Dim=30 | | Dim=50 | |
|---|---|---|---|---|---|---|
| | 3Sources | Cora | 3Sources | Cora | 3Sources | Cora |
| CCA | 64.72 | 44.17 | 71.52 | 48.37 | 80.51 | 51.24 |
| MSE | 65.31 | 46.28 | 70.96 | 48.85 | 80.34 | 50.22 |
| FLSSS | 63.28 | 47.12 | 72.58 | 47.31 | 77.71 | 46.59 |
| MISL | 70.29 | 50.21 | 79.93 | 53.26 | 86.28 | 56.38 |
| McDR | 69.38 | 48.11 | 76.08 | 51.18 | 82.68 | 52.76 |
| U-MVML-LA | 71.55 | 47.63 | 79.12 | 51.08 | 85.92 | 55.28 |
| **MvSLGE (Ours)** | **81.27** | **55.26** | **86.98** | **60.71** | **89.93** | **62.45** |

framework to integrate multi-view features into one common latent representation and preserve local geometry structure. This is because our proposed MvSLGE can fully consider the local structure when integrating the complementary information of multi-view features to construct the ideal latent representation.

Then, we conduct image classification experiments on 3 widely used image datasets: Caltech 101-7, MSRC-v1, and COIL-20 datasets.The experimental setting is same to the experiment on text datasets. To evaluate the robustness of MvSLGE, the training images of all datasets are corrupted with different strengths. For example, 10% corrupted strength means randomly selecting 10% pixels of the image and replacing these pixels with 0. We select one image of COIL-20 and demonstrate the original image and corrupted image in Figure 3.

For the Caltech101-7 dataset, all the images are represented in terms of 6 views features: Gabor, WM, CENTRIST, HOG, GIST, and LBP. MSRC-v1 consists of 240 images from 9 categories. In our experiments, we utilize 210 samples from 7 classes and extract 6 view features: GIST, HOG, LBP, CENT, SIFT, and CMT. For COIL-20 dataset, we extract the features from 3 views: intensity (view1), LBP (view2), and Gabor (view3). The classification accuracies of all comparison methods are shown in Table 3.

Through Table 3, it can be clearly seen that MvSLGE achieves the best results in most situations. Because Cauchy loss adopted by MISL can reduce the influence of outliers, MISL also achieves promising results. These results illustrate that incorporating the local geometry information with complementary information is effective for learning an ideal latent representation.

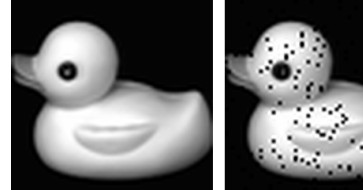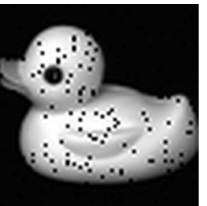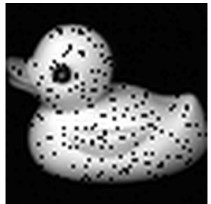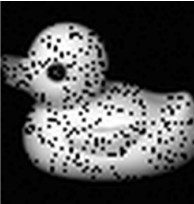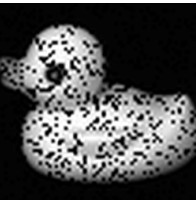

Figure 3: From left to right: the original image in COIL-20 dataset, image with 5% corruption, image with 10% corruption, image with 15% corruption, image with 20% corruption.

Table 3: The accuracy of the classification and robust testing on Caltech 101-7, MSRC-v1 and COIL20 datasets.

| Datasets | Corruption | 0% | 5% | 10% | 15% | 20% |
|---|---|---|---|---|---|---|
| Caltech101-7 | CCA | 83.41 | 79.36 | 76.43 | 75.18 | 73.11 |
| | MSE | 82.86 | 80.42 | 79.88 | 76.12 | 75.27 |
| | FLSSS | 84.00 | 83.08 | 81.73 | 78.11 | 76.30 |
| | MISL | 86.58 | 85.34 | 84.21 | 82.15 | 81.98 |
| | McDR | 85.19 | 83.29 | 81.27 | 80.22 | 78.36 |
| | U-MVML-LA | 88.76 | 86.04 | 84.33 | 83.45 | 80.23 |
| | **MvSLGE (Ours)** | **91.31** | **89.28** | **88.65** | **87.74** | **86.22** |
| MSRC-v1 | CCA | 80.53 | 71.92 | 72.01 | 80.88 | 77.25 |
| | MSE | 86.16 | 83.25 | 83.01 | 81.34 | 79.88 |
| | FLSSS | 84.11 | 83.29 | 80.52 | 80.27 | 78.98 |
| | MISL | 86.29 | 86.11 | 86.04 | 85.28 | 84.89 |
| | McDR | 87.71 | 86.29 | 85.98 | 85.31 | 84.22 |
| | U-MVML-LA | 85.06 | 84.86 | 84.23 | 83.68 | 83.03 |
| | **MvSLGE (Ours)** | **88.76** | **87.95** | **87.24** | **86.89** | **86.51** |
| COIL20 | CCA | 83.53 | 77.36 | 68.93 | 62.22 | 56.21 |
| | MSE | 85.86 | 76.61 | 69.37 | 63.02 | 59.93 |
| | FLSSS | 83.29 | 74.27 | 66.19 | 60.82 | 52.46 |
| | MISL | 90.58 | 85.22 | 84.86 | 81.13 | 78.15 |
| | McDR | 88.98 | 76.16 | 68.35 | 57.83 | 55.58 |
| | U-MVML-LA | 91.76 | 84.64 | 81.19 | 79.17 | 76.84 |
| | **MvSLGE (Ours)** | **93.17** | **87.16** | **85.97** | **82.95** | **82.12** |

## 4.4 COMPARISON WITH DEEP LEARNING METHOD

Deep learning as a branch of machine learning is based on artificial neural networks to learn data representations. In this section, we compare our proposed algorithm with two famous deep learning algorithms ResNet He et al. (2016) and Vision Transformer (VIT) Dosovitskiy et al. (2020) for the image classification task. In this experiment, we exploit ResNet-34 and VIT_B_16 to perform the image classification task. To compare with deep learning algorithms, we utilize one additional large-scale image dataset MINIST combining with previous three datasets to do the experiments. For MINIST, we utilize grayscale intensity (GSI), LBP, and Gabor as multi-view features. The performances of classification accuracy on 4 datasets are shown in Table 4. From Table 4 we can find that the classification performances of our MvSLGE and deep learning algorithms are comparable. But the advantage of our MvSLGE is that it is the unsupervised method and does not relate to the training which is fast for the implementation, while other deep learning algorithms are supervised methods which need to employ the labels to extract discriminative information for constructing the representation.

## 4.5 DISCUSSION

In this section, we conduct the classification on three image datasets to show the influence of the auto-weighted scheme. We fix all weights $a_v$ as $\frac{1}{m}$ and show the experimental results in Table 5. We can see from Table 5 that the auto-weighted scheme can effectively improve the performance of

Table 4: The comparison of the classification performances of the proposed algorithm with ResNet-34 and VIT_B_16 in for datasets.

| Method | MINIST | Caltech 101-7 | MSRC | COIL-20 |
|---|---|---|---|---|
| ResNet-34 | 97.93 | **95.54** | **94.73** | 95.61 |
| VIT_B_16 | **98.82** | 94.69 | 92.23 | **97.58** |
| MvSLGE (Ours) | 91.18 | 91.31 | 88.76 | 93.17 |

Table 5: The influence of the proposed auto-weighted scheme.

| | Datasets | | |
|---|---|---|---|
| | Caltech 101-7 | MSRC-v1 | COIL-20 |
| $a_v = \frac{1}{m}$ | 89.19 | 87.75 | 85.82 |
| Flexible $a_v$ | **91.31** | **88.76** | **93.17** |

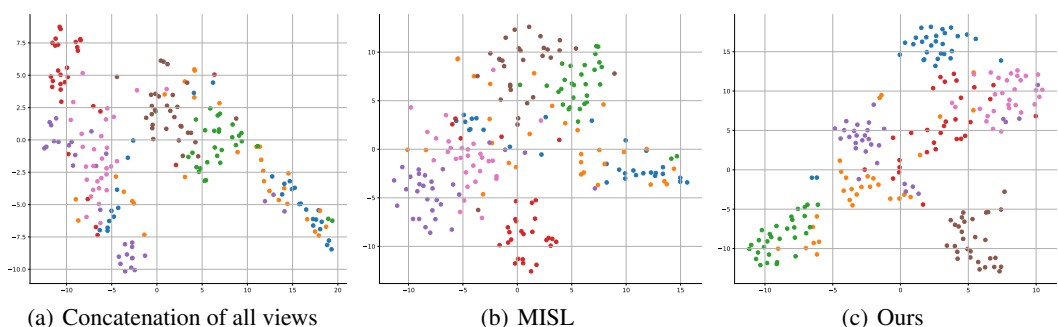

(a) Concatenation of all views        (b) MISL        (c) Ours

Figure 4: Figure (a) corresponds to the concatenation of the features of all views. Figure (b) and Figure (c) are the latent representations obtained by MISL and MvSLGE respectively.

MvSLGE to obtain better results than simply setting all the weights to $\frac{1}{m}$. This result demonstrates the effectiveness of the auto-weighted scheme for MvSLGE.

Furthermore, we show the visualization of the distribution of the latent representations learned from MvSLGE on MSRC-v1 dataset. The t-SNE Van der Maaten & Hinton (2008) is exploited to embed the concatenation of all views features and the learned latent representations into a 2-dimensional subspace to demonstrate the distributions in Figure 4. We clearly find that MISL and MvSLGE can reveal the underlying structure better than the original feature. Especially, by the latent representation learned from MvSLGE, the samples can be separated into more compact clusters. More discussion of the experiments for the parameters setting, analysis of convergence, and visualization results are shown in the section 5 of supplemental material.

## 5 CONCLUSION

In this work, we introduce a novel latent space learning model MvSLGE for multi-view data. MvSLGE aims to construct a latent representation by incorporating complementary information from multiview features to comprehensively depict each sample. Moreover, MvSLGE employs an adaptively learned affinity matrix, which well characterizes the relationship between samples, to regularize the latent representation. In the process of learning the affinity matrix, MvSLGE integrates both the local manifold structure of original multi-view features and the latent representations. Therefore, by combining latent representation learning, manifold embedding, and graph learning into one framework, our approach can further prefer to extract the local manifold structure from multi-view features and encode it into the latent representations to improve the discriminability. Various experiments have demonstrated that the proposed MvSLGE is an effective multi-view subspace learning method. In the future, some accelerating techniques will be considered in the framework to deal with large scale data, and nonlinearity by kernel or neuro networks will be introduced into our framework.

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
