# A Multi-view Latent Space Learning Framework via Adaptive Graph Embedding

## 1. OPTIMIZATION

In this section, we demonstrate the rest parts of the optimization, which contains the steps of updating $S$, $a$ and $E$, and the details of processing test sample.

(3). $S$-subproblem: In the step of updating affinity matrix $S$, we fix $X$, $\mathcal{E}$, $a$ and $\mathcal{P}$, and obtain the following problem

$$
\begin{aligned}
S^* &= \arg\min_S \eta \sum_{i,j=1}^N \left\| x_i - x_j \right\|_2^2 s_{ij} + \zeta \sum_{v=1}^m a_v \left\| S - S^{(v)} \right\|_F^2 \\
&= \arg\min_S \sum_{i=1}^N \left( \eta s_i h_i^T + \zeta \sum_{v=1}^m a_v \left\| s_i - s_i^{(v)} \right\|_2^2 \right), \\
&\text{s.t.} \quad s_{ij} \geq 0 \quad \text{and} \quad \sum_j s_{ij} = 1,
\end{aligned}
\tag{S1}
$$

where $h_i$ is a row vector with $h_i(j) = \left\| x_i - x_j \right\|_2^2$. Obviously, Equation S1 is independent of each row of $S$. Therefore, we can optimize each $i$-row of $S$ individually with ignoring the constant term

$$
\begin{aligned}
&\min_{s_i} \zeta s_i s_i^T + s_i \hat{h}_i^T, \\
&\text{s.t.} \quad s_{ij} \geq 0 \quad \text{and} \quad \sum_j s_{ij} = 1,
\end{aligned}
\tag{S2}
$$

where $\hat{h}_i^T = \eta h_i - 2\zeta \sum_{v=1}^m s_i^{(v)}$, and $s_i^{(v)}$ is the $i$-th row of $S^{(v)}$. Equation S2 can be effectively solved by [1].

(4). $a$-subproblem: With $P$, $E$, $X$ and $S$ fixed, we can update the weights for all views by the following problem

$$
\begin{aligned}
&\min_a \sum_{v=1}^m a_v \left\| S - S^{(v)} \right\|_F^2 + \gamma ||a||_2^2, \\
&\text{s.t.} \quad a_v \geq 0 \quad \text{and} \quad \sum_v a_v = 1.
\end{aligned}
\tag{S3}
$$

We can easily find that the optimal solution of Equation S3 can be effectively obtained by solving the complementary slackness KKT condition.

(5). $E$-subproblem: With $P$, $X$, $S$ and $a$ fixed, we can update the error $E$ for each view by the following problem

$$
\begin{aligned}
E^* &= \arg\min_E \|E\|_{2,1} + \mathcal{G}\left( J, Z - PX - E \right), \\
&= \arg\min_E \frac{1}{\mu} \|E\|_{2,1} + \left\| E - (Z - PX + J/\mu) \right\|_F^2.
\end{aligned}
\tag{S4}
$$

Equation S4 can be solved efficiently by a lemma in [2]

**Lemma**: Let $Q = [q_1, q_2, ..., q_n]$ be a given matrix and $\|\cdot\|_F$ be the Frosenius norm. If the optimal solution of

$$
\min_W \alpha \|W\|_{2,1} + \frac{1}{2} \|W - Q\|_F^2,
\tag{S5}
$$

is $W^*$, then the $i$th column of $W^*$ is

$$
W^*(:,i) = \begin{cases} \dfrac{\|q_i\| - \alpha}{\|q_i\|} q_i, & \text{if} \quad \alpha < \|q_i\| \\ 0, & \text{otherwise.} \end{cases}
\tag{S6}
$$

(6). **Updating Multiplier**: We update the multiplier by

$$J = J + \mu \left( Z - PX - E \right). \tag{S7}$$

We iteratively alternate the five steps until the objective function satisfies convergence conditions.

In the testing stage, for a new sample with the multi-view features $\left\{ z^{(1)}, z^{(2)}, ..., z^{(v)} \right\}$, we can construct the latent representation $x$ of the sample by the following objective function

$$\min_{x} \quad \frac{1}{m} \sum_{v=1}^{m} \left\| z^{(v)} - P_v^* x \right\|_2 + \lambda \left\| x \right\|_2^2, \tag{S8}$$

where $P_v^*$ is the obtained optimal view generation from training stage. We adopt the iteratively reweight (IR) technique to solve the above problem.

We first compute the gradient of Equation S8 with respect to $x$ and set it to 0

$$\sum_{v=1}^{m} - \frac{2 \left( P_v^* \right)^T \left( z^{(v)} - P_v^* x \right)}{\left\| z^{(v)} - P_v^* x \right\|_2} + 2m\lambda x = \mathbf{0}, \tag{S9}$$

which can be further rewritten as

$$\left( \sum_{v=1}^{m} \frac{\left( P_v^* \right)^T P_v^*}{\left\| z^{(v)} - P_v^* x \right\|_2} + 2m\lambda I \right) x = \sum_{v=1}^{m} \frac{\left( P_v^* \right)^T z^{(v)}}{\left\| z^{(v)} - P_v^* x \right\|_2}, \tag{S10}$$

where the residual of $x$ on the $v$th view is set as $r^{(v)} = z^{(v)} - P_v^* x$. We can define the weights

$$R = \left[ \frac{1}{\left\| r^{(1)} \right\|_2 + \epsilon}, \frac{1}{\left\| r^{(2)} \right\|_2 + \epsilon}, ..., \frac{1}{\left\| r^{(m)} \right\|_2 + \epsilon} \right], \tag{S11}$$

where $\epsilon > 0$ is a very small value to avoid dividing by zero. Based on Equation S10 and Equation S11, we have

$$x = \left( \sum_{v=1}^{m} R_v \left( P_v^* \right)^T P_v^* + 2m\lambda I \right)^{-1} \sum_{v=1}^{m} \left( P_v^* \right)^T R_v z^{(v)}, \tag{S12}$$

where $R_v = \frac{1}{\left\| r^{(v)} \right\|_2 + \epsilon}$. It can be seen that the weights can effectively reduce the influence of outliers. Moreover, $\mathcal{J}$ is a convex function that has a globally optimal solution. Thus, we can iteratively update the latent representation $x$ using Equation S12 with any initial estimate until convergence.

## 2. THE DETAILS OF THE PROCEDURE

In Table S1, We summarize the procedure of our MvSLGE, which contains the details of initialization of parameters and the convergence conditions.

**Table S1.** The procedure of MvSLGE

**Input:**

Original multi-view data: $\left\{ \mathbf{Z}^{(1)}, \mathbf{Z}^{(2)}, ..., \mathbf{Z}^{(m)} \right\}$;

Latent representation $x$ dimensionality: $D$ and hyperparameters: $\lambda, \eta, \gamma$ .

**Initialization:**

Initializing similarity matrix $S$ by solving $\min_{s_i \mathbf{1}=1, s_{ij} \geq 0} \sum_{v=1}^{m} \frac{1}{m} \left\| s_i - s_i^{(v)} \right\|_2^2$

$\mathbf{P} = \mathbf{0}, \mathbf{J} = \mathbf{0}, \mathbf{E} = \mathbf{0}, \mu = 10^{-6}, \epsilon = 10^{-3}, \rho = 1.2, \max_\mu = 10^6$;

Initializing $X$ with random values;

**The optimization procedure of MvSLGE**:

While not converged, do:

    According to subproblems 1-4, update variables $\mathbf{P}_v, \mathbf{X}, \mathbf{S}, \mathbf{E}^{(v)}$;

    According to Equation S7 update multiplier $\mathbf{J}$ ;

    Updating $\mu$ by $\mu = \min(\rho\mu; \max_\mu)$;

Until satisfying convergence condition:

    $\|\mathbf{Z} - \mathbf{PX} - \mathbf{E}\|_\infty < \epsilon.$

**Output:**

The view generation functions: $\{\mathbf{P}_v\}_{v=1}^{m}$, and latent representation: $\mathbf{X}$.

## 3. COMPLEXITY

The optimization procedure of MvSLGE consists of 6 subproblems. We set $m$, $N$, $D$, and $d$ as the numbers of views and samples, the sum of dimensionality of all views features, and the dimensionality of the learned latent representation. The complexity of updating the view generation functions $\{\mathbf{P}_v\}_{v=1}^{m}$ is $O\left(d^2 D + D^3\right)$. For updating the latent representation $\mathbf{X}$ by Bartels-Stewart algorithm, the complexity is $O\left(d^3\right)$. For updating the graph $\mathbf{S}$, the complexity is $O\left(dN^2 + Nd^2\right)$. For updating the weights $\mathbf{a}$, the complexity is $O\left(N^3\right)$. The complexity of updating the error $\mathbf{E}$ and multipliers is $O\left(dDN + DN^2\right)$. Therefore, the complexity of MvSLGE is $O\left(d^2 D + D^3 + d^3 + dN^2 + Nd^2 + N^3 + dDN + DN^2\right)$. Since $d \ll D$, the complexity can be simplified as $O\left(D^3 + N^3\right)$.

## 4. THE DETAILS OF DATASETS

All the experiments are conducted on 7 famous datasets: the Holidays and Outdoor Scene are utilized for image retrieval, while the 3Sources, Cora Caltech 101-7, MSRC-v1, and COIL-20 are utilized for classification experiments. We summarize the details of the employed datasets in Table S2.

**Table S2.** The detailed information of all datasets.

| Datasets | Sizes | Classes | Views |
|---|---|---|---|
| Holidays | 1491 | 500 | 3 |
| Outdoor Scene | 2688 | 8 | 4 |
| Caltech101-7 | 441 | 7 | 6 |
| MSRC-v1 | 240 | 9 | 6 |
| COIL-20 | 1440 | 20 | 3 |
| 3Sources | 169 | 6 | 3 |
| Cora | 2708 | 7 | 2 |

## 5. THE DISCUSSION OF EXPERIMENTS

In this section, we further evaluate factors influencing the performance of MvSLGE and discuss the convergence of MvSLGE. The image classification experiments are conducted to discuss the effect of the hyperparameter $\eta$ and $\zeta$ on the performance of MvSLGE. We take the classification experiments by setting different values of $\eta$. The performance of MvSLGE is shown in Figure S1.

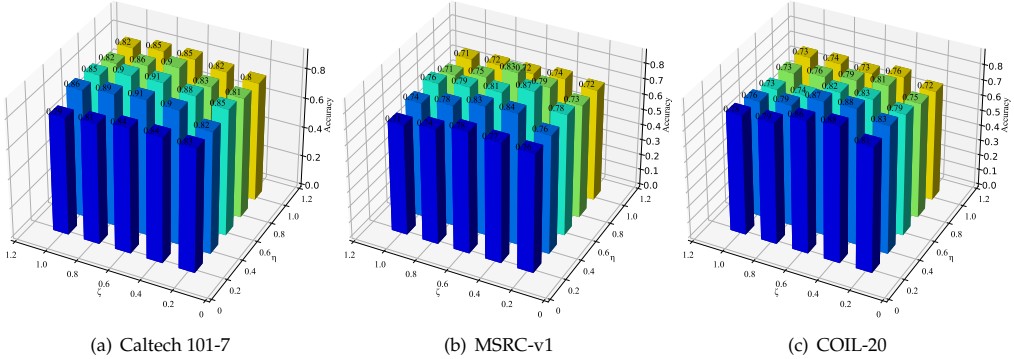

| (a) Caltech 101-7 | (b) MSRC-v1 | (c) COIL-20 |
|---|---|---|

**Fig. S1.** Results of the proposed with different values of $\eta$ and $\zeta$.

The optimization procedure of MvSLGE consists of 6 subproblems. Generally proving the convergence of the algorithm is difficult. However, the promising results on various datasets show that MvSLGE has stable convergence with the latent representation initializing randomly. In Figure S2, we show the curve about values of the objective function and iteration numbers, to empirically demonstrate the convergence of our algorithm. It can be seen from Figure S2 that our algorithm converges in 10 times iterations in general.

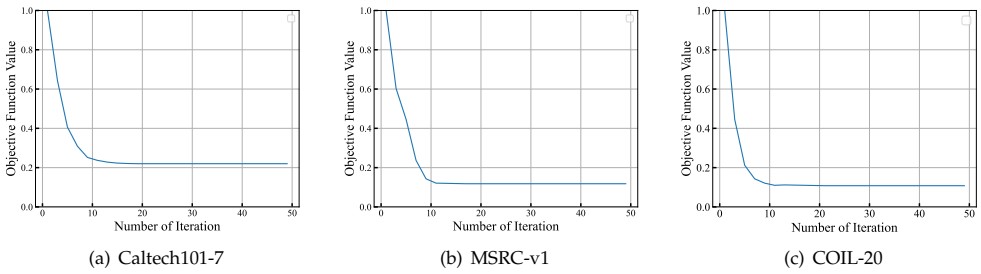

| (a) Caltech101-7 | (b) MSRC-v1 | (c) COIL-20 |
|---|---|---|

**Fig. S2.** The values of the objective function with the numbers of iterations on 3 image datasets. All plots are normalized into $[0, 1]$.

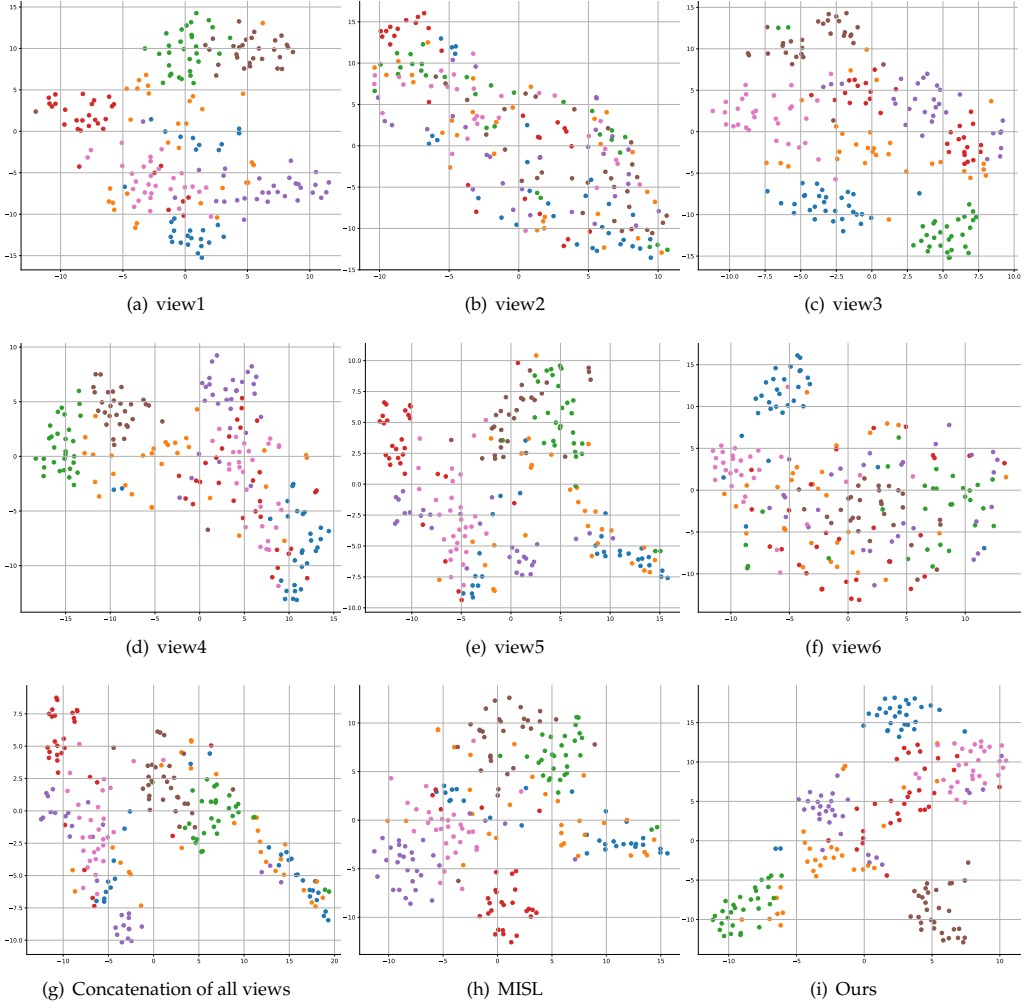

(a) view1      (b) view2      (c) view3

(d) view4      (e) view5      (f) view6

(g) Concatenation of all views      (h) MISL      (i) Ours

**Fig. S3.** Visualization of the representations of samples on MSRC-v1 dataset with t-Distributed Stochastic Neighbor Embedding (t-SNE). Figure (a)-Figure (f) is the visualization of features from each view. Figure (g) corresponds to the concatenation of the features of all views. Figure (h) and Figure (i) are the latent representations obtained by MISL and MvSLG respectively.

Furthermore, we show the visualization of the distribution of the latent representations learned by MvSLG on MSRC-v1 dataset. Beforehand, we utilize MISL and MvSLG to construct the 200-dimensional latent representations from multi-view features. The t-SNE is exploited to embed all the high-dimensional original multi-view features and the learned latent representations into a 2-dimensional subspace to demonstrate the distributions in Figure S3.

As shown in Figure S3, although the sample distributions of some classes obtained from the original multi-view feature are relatively compact, most sample distributions are disordered. We can clearly find that MISL and MvSLG can reveal the underlying structure better than the original feature. Especially, by the latent representation learned from MvSLG, the samples can be separated into more compact clusters.