# OpenReview forum: "A multi-view latent space learning framework via adaptive graph embedding"
_ICLR.cc/2024/Conference — ICLR 2024 Conference Withdrawn Submission_

### Official Review · Reviewer_yhSb · 2023-10-21

**Soundness:** 2 fair
**Presentation:** 2 fair
**Contribution:** 2 fair
**Rating:** 5
**Confidence:** 4

**Summary:**

This paper proposes a new approach to multi-view subspace learning termed as multi-view latent space learning via adaptive graph embedding (MvSLGE), which learns a latent representation from all view features. Unlike most existing multi-view latent space learning methods that only encode the complementary information into the latent representation, MvSLGE adaptively learn an adjacent graph that well characterizes similarity between samples to further regularize the latent representation. To extract the neighborhood information from multi-view features, the authors propose a novel strategy that constructs one graph for each view, and then the learned graph is approximately designed as a centroid of these graphs of different views with different weights.

**Strengths:**

1. The originality, quality, and significance of this paper is supported by the proposed MvSLGE, which fuses the multi-view features to construct one common latent representation and incorporate complementary information of heterogeneous property features.

2. The clarity of this paper is satisfied based on the clear listed motivations and contributions of their work.

**Weaknesses:**

1. The biggest problem of this paper is the limited novelty in formulation shown in Eq. (5), which adopts different regulation terms to consider the outliers, Laplacian embedding and local geometry structure from multi-view features. The combination of these regulation terms more or less limits the novelty of this paper, which seems to be incremental compared with the existing works. I hope the authors can present their work from a higher perspective, even based on the combination of different regulation terms, i.e., the related theoretical illustration or other aspects.

2. The performance improvements in this paper are obtained based on the added several regulation terms, which more or less adds the computation cost compared with those methods without these several regulation terms. I hope the authors can analyze the computation cost at the end of section 3.2.

3. The compared methods in classification and robust testing in the experiment are not enough. The authors are expected to add more representative or the latest methods for comparison in the experiment. Besides, the datasets adopted for comparison are not enough.

4. The authors are expected to repeat each experiment for many times and record the mean and devition to avoid the possible randomness in the experiment, i.e., Table 2 and Table 3.

5.  The ablation study regarding the added three regulation terms are all expected to conduct in the experiment and just investigate the auto-weighted scheme is not enough.

6. The authors are suggested to adopt other symbol to represent the common latent representation rather than X in this paper. As is known, X is usually adopted the original dataset and this is the first time for me to see that X is adopted to represent the others in this area.

**Questions:**

1. Apart from corruption, have the authors thought whether the proposed method still performs well when there is missing samples for the dataset. Which one performs better in the experiment, corruption or missing cases?

---

### Official Review · Reviewer_kTLj · 2023-10-28

**Soundness:** 3 good
**Presentation:** 3 good
**Contribution:** 3 good
**Rating:** 1
**Confidence:** 5

**Summary:**

This paper investigates the multi-view subspace clustering problem by using multi-view graph fusion. As the authors claimed, the overall idea of this paper lies in using single graph learning on each view followed by multi-view graph integration. Some experiments validate the performance of the proposed method when compared to other competing algorithms.

**Strengths:**

Robust feature learning with L21 norm is used to improve the efficacy against the original raw features. Locality-preserving graph is used to formulate the graph structure on each view. A simple common graph learning is used to produce a consensus graph. Experiences on several datasets show the effectiveness of the proposed method.

**Weaknesses:**

The novelty of this paper is out of date and incremental to multi-view clustering as well as multi-view learning. The overall idea of this paper simply combines the Belkin & Niyogi (2001) for graph learning,  the objective function Xu et al. (2015), and others.

The optimization process is built on the well-known ADMM, which is not the main technical contribution.

The given datasets are commonly used for many works for multi-view clustering validation.

For the performance, for all these datasets, if we take deep multi-view clustering into consideration, how about the performance comparison? It is clear the clustering results are far from satisfactory.

**Questions:**

The technical contribution of this paper is trivial and the reviewer could not find their potential in real-world applications.

---

> ### Comment · Reviewer_kTLj · 2023-11-22
>
> After reading the comments from other reviewer, the authors did not provide any feedback. 'reject' is our final recommendation.

---

### Official Review · Reviewer_F9J3 · 2023-10-30

**Soundness:** 2 fair
**Presentation:** 2 fair
**Contribution:** 2 fair
**Rating:** 3
**Confidence:** 4

**Summary:**

The authors proposed a multi-view algorithm called MvSLGE, which establishes an adaptative graph combining these graphs of all views to explore the local structure of data in the subspace. In addition, use $\ell_{2,1}$-norm constraint on reconstruction loss to avoid outliers. Experiments are conducted to validate the performance of the proposed method.

**Strengths:**

The authors conduct sufficient experiments to validate the performance of the proposed method.

**Weaknesses:**

1. The novelty of this paper is limited.
2. The comparison of Tabel 1 should be extended to state-of-the-art methods. Comparing algorithms are too old.

**Questions:**

1. The novelty of this paper is limited. The author's paper does not meet the standards for publication on ICLR.
2. In the experiment, the authors should add a comparison with some state-of-the-art methods in the past three years. Such a comparison will be convincing.
3. The constraint on similarity graph S in Eq(5)  is s_ij >0. In contrast, s_ij in Eq(6) is non-negative. The authors need to check it carefully.
4. In the related work section, more literature should be considered.
5. The paper writing needs improvements.